# Automatic Emotion Recognition from EEG Signals Using a Combination of Type-2 Fuzzy and Deep Convolutional Networks

**Farzad Baradaran [1], Ali Farzan [1,\*], Sebelan Danishvar [2,\*] and Sobhan Sheykhivand [3]**

[1] Department of Computer Engineering, Shabestar Branch, Islamic Azad University, Shabestar 5381637181, Iran
[2] College of Engineering, Design and Physical Sciences, Brunel University London, Uxbridge UB8 3PH, UK
[3] Department of Biomedical Engineering, University of Bonab, Bonab 5551395133, Iran
**\*** Correspondence: alifarzan402@gmail.com (A.F.); sebelan.danishvar@brunel.ac.uk (S.D.)

**Abstract:** Emotions are an inextricably linked component of human life. Automatic emotion recognition can be widely used in brain–computer interfaces. This study presents a new model for automatic emotion recognition from electroencephalography signals based on a combination of deep learning and fuzzy networks, which can recognize two different emotions: positive, and negative. To accomplish this, a standard database based on musical stimulation using EEG signals was compiled. Then, to deal with the phenomenon of overfitting, generative adversarial networks were used to augment the data. The generative adversarial network output is fed into the proposed model, which is based on improved deep convolutional networks with type-2 fuzzy activation functions. Finally, in two separate class, two positive and two negative emotions were classified. In the classification of the two classes, the proposed model achieved an accuracy of more than 98%. In addition, when compared to previous studies, the proposed model performed well and can be used in future brain–computer interface applications.

**Keywords:** emotion recognition; deep learning; electroencephalography; generative adversarial networks

## 1. Introduction

Computers will become an increasingly important part of human life in the near future. They are, however, emotionally blind and cannot comprehend human emotional states [1]. Human–computer interaction (HCI) performance can be improved by reading and understanding human emotional states [2]. As a result, the exchange of this information and recognition of the user's affective states are deemed necessary in order to improve human–computer interaction. An emotion recognition system can be useful in a variety of applications, including intelligent learning, computer games, and health applications [3]. The emotional state of a person can be detected using physiological indicators such as EEG, electrocardiography (ECG), electrical skin response, and facial signs [4]. However, because of its sensitivity to social coverage, diagnosis based on physiological indicators is less considered and used. Electroencephalography (EEG) is the most popular and widely used physiological signal for detecting various emotions [5]. Evidence suggests that there is a strong relationship between this signal and emotions such as happiness, sadness, anger, and so on. EEG electrodes are placed on the head to record brain electrical activity [6]. Because EEG signals are directly derived from the central nervous system (CNS) and exhibit characteristics related to internal emotional states, they have a strong association with various emotions [7]. As a result, by utilizing these non-invasive technologies, we may be able to develop an emotion recognition system that can be used in everyday life [8]. This study focuses on developing emotion classification models using EEG signals and machine learning techniques.

Emotions are formed by one's thoughts. Because emotions are expressed as reactions to external stimuli when people think, communicate, learn, and make decisions, they play a critical role in determining people's behavior in everyday life. Emotions, on the other hand, are a completely subjective phenomenon because different people in the same environment and environmental conditions react differently to external stimuli. All human emotions are derived from a few fundamental emotions. Some researchers believe that a wide range of complex emotions can be classified into six states. Anger, disgust, fear, joy, sadness, and surprise are among these emotions. These emotions are expressed separately and are unrelated to one another. It is not always possible to accurately describe feelings by using only a few emotions. Psychologists use factor analysis techniques to measure the correlation between emotions in order to accurately diagnose different emotions. The three main dimensions of capacity, arousal, and power are thus defined, from which emotions can be described [8,9].

Stimuli cause emotions, which can then be measured and used in applications such as brain–computer interfaces (BCI), lie detection, and so on. Stimuli are classified into two types: induced stimuli and naturalistic stimuli. Induced emotions are those caused by stimuli that are purposefully chosen to elicit different emotions in people, such as movies, images, and music. Naturalistic stimuli, on the other hand, refer to uncontrollable natural situations and stimuli. There are numerous limitations to the naturalistic stimulus for emotion recognition, and these emotions are beyond the scope of this study, due to the difficulties in collecting and annotating them in the real world. In the field of emotion recognition research, induced stimuli are very popular. In recent studies, various stimuli such as events, images [10], music [11], or movies [12] have been used to elicit emotions. Music was used to stimulate emotions in this study because music is very effective at expressing emotions and evoking emotional responses in the listener. Music is known as the language of emotions and is recognized as a powerful method that can elicit various emotional reactions in the listener such as happiness, excitement, and fear [1–4]. When a person is unhappy, listening to happy music can help lift their spirits. When people listen to music, it is extremely rare that they do not experience some level of emotion.

One of the difficulties in emotion recognition is determining the truth of the emotion stimulus's test context [13] through annotation or interpretation. Obviously, determining the ground truth of emotions is difficult because there is no clear definition of emotions, but the best way to determine or interpret emotions during an experiment is for the test subject to subjectively rate the emotional trials or report them. The Self-Assessment Manikin (SAM) is one of these tools designed to assess people's emotional experience [13]. Many studies have been conducted in order to detect emotions from EEG signals automatically. In the following, each of the recent studies for the automatic detection of emotions will be examined, as well as the advantages and disadvantages of each study.

Li et al. [14] presented a new model to automatically recognize emotion from EEG signals. These researchers used 128 channels to record the EEG signal. In addition, they identified active channels in 32 participants using correlation-based feature selection (CFS) and reported that five channels, T3, F3, Fp1, O2, and Fp2 have a great effect on emotion recognition. These researchers used the genetic algorithm (GA) to reduce the dimensions of the feature vector and used the *t*-test to verify the correctness of the selected features. Finally, the k-nearest neighbors (KNN), random forest (RF), Bayesian, and multilayer perceptron (MLP) classifiers were used for classification. Yimin et al. [15] recognized the four emotions of happiness, sadness, surprise, and peace from EEG signals. They used eight volunteers to record the EEG signal. These researchers used four classifiers, RF, linear discriminant analysis (LDA), support vector machine (SVM), and C4.5 for the classification part and concluded that the C4.5 classifier has a better performance in detecting emotion. Hassanzadeh et al. [16] used a fuzzy parallel cascade (FPC) model to detect emotion. For their experiment, these researchers used a musical stimulus with 15 participants. They also compared their proposed model with recurrent neural networks (RNN). Finally, the mean squared error (MSE) of these researchers for the classification of the two classes of

valence and arousal is reported to be around 0.089, which is lower compared to other models. Panayo et al. [17] used deep neural networks (DNNs) to recognize emotion from EEG signals. They conducted their experiment on 12 people. Their proposed network architecture consisted of six convolutional layers. In addition, these researchers compared their proposed algorithm with SVM and concluded that the convolutional neural network (CNN) has better performance in emotion recognition than comparative methods. Chen et al. [18] used EEG signals to automatically classify two classes of emotion. These researchers used parallel RNNs in their proposed algorithm. The final reported accuracy for the valence- and arousal-class classification based on their proposed algorithm was reported as 93.64% and 93.26%, respectively. He et al. [19] used dual wavelet transform (WT) to extract features from EEG signals in order to recognize emotion. In addition, these researchers, after feature extraction, used recursive units to train their model. Finally, they achieved an accuracy of 85%, 84%, and 87% for positive, negative, and neutral emotion classes, respectively. Sheykhivand et al. [20] used 12 channels of EEG signals for the automatic recognition of emotion. For this purpose, these researchers used a combination of RNN and CNN for feature selection/extraction and classification. In their proposed model, they identified three different states of emotion, including positive, negative, and neutral, using musical stimulation, and achieved 96% accuracy. Among the advantages of their model, the classification accuracy is high, and the computational complexity can be considered as the disadvantage of this research. Er et al. [21] presented a new model for automatic emotion recognition from EEG signals. These researchers used transfer-learning networks such as VGG and AlexNet in their proposed model. They achieved satisfactory results based on the VGG network in order to classify four different basic emotional classes, including happy, relax, sad, and angry. Despite the advantages of this research, low computational complexity and low classification accuracy can be considered as disadvantages of this research. Gao et al. [22] presented a new model for automatic emotion recognition. Their model consisted of two different parts. The first part consisted of a novel multi-feature fusion network that used spatiotemporal neural network structures to learn spatiotemporal, distinct emotional information for emotion recognition. In this network, two common types of features, time domain features (differential entropy, sample entropy) and frequency domain features (power spectral density) were extracted. Then, in the second part, they were classified into different classes using Softmax and SVM. These researchers used the DEAP dataset to evaluate their proposed model, and achieved promising results. However, computational complexity can be considered a disadvantage of this research. Hou et al. [23] used the feature pyramid network (FPN) to improve emotion recognition performance based on EEG signals. In their proposed model, the differential entropy (DE) of each recorded EEG channel was extracted as the main feature. These researchers used SVM to score each class. The accuracy reported by these researchers in order to detect the dimensions of valence and arousal for the DEAP database was reported as 94% and 96%, respectively. Among the advantages of this research is high classification accuracy. In addition, due to the computational complexity, their proposed model cannot be implemented on real-time systems, which can be considered a disadvantage of this research.

Previous studies, as previously discussed, have several fundamental limitations. The absence of a comprehensive public database based on musical stimulation of EEG signals is the first limitation of previous studies. There is almost no comprehensive database for emotion recognition based on musical stimulation based on EEG signals, and the majority of databases examined in previous studies are private. The use of manual feature-extraction/selection algorithms is the second limitation of previous research. Most previous studies used engineering and manual feature-selection/extraction methods, which does not guarantee the optimality of the obtained feature vector for each subject/problem, and this issue can cause a decrease in classification accuracy and an increase in computational efficiency. Deep learning networks have been used in some recent studies to propose models. However, due to their computational complexity, these studies are not suitable for use in real-time applications, which is the third limitation of previous research. The goal of

this research is to overcome the limitations mentioned above and provide an efficient model for automatic emotion recognition. To begin, a standard and publicly available database of EEG signals based on musical stimulation is compiled. The data is then fed into generative adversarial networks (GAN) to avoid overfitting. After the data is increased, the samples are entered into the proposed model, which is based on a combination of deep CNN and type-2 fuzzy networks, and are classified into positive and negative classes. This work's contribution can be summarized as follows:

- Compilation of a standard database for automatic emotion recognition from EEG signals using musical stimulation.
- Providing an end-to-end algorithm for automatically detecting emotions from EEG signals without the use of feature selection/extraction block diagrams.
- Providing an automatic method for emotion recognition from EEG signals that is based on a combination of GAN, CNN, and type-2 fuzzy networks, as well as a customized architecture with high speed and accuracy.
- Providing an automatic method for emotion recognition from EEG signals that can withstand a wide range of environmental noises.

The remainder of the paper is structured as follows: Section 2 delves into the mathematical foundations of GAN, CNN, and type-2 fuzzy networks. The proposed research model, including the method of data collection, data pre-processing, and the details of the proposed architecture, is then examined in Section 3. Section 4 looks at the simulation results and compares them to previous research. Section 5 is finally related to the conclusion.

## 2. Materials and Methods

The mathematical background of GAN, CNN, and type-2 fuzzy networks will be examined in this section, and relevant details about these networks will be presented.

### 2.1. Brief Description of GAN

GANs have grown in popularity in the field of deep learning over the last few years. The GANs consist of two main discriminator (D) and generator (G) networks. These two components are acting exactly opposite each other. The first network, the discriminator, is trained to distinguish between real and fake input data. The second network, the generator, takes a latent noise variable z as input and tries to generate fake samples that are not recognized as fake by the discriminator. The D is in charge of telling the difference between real and fake signals. A network D is trained to distinguish between original and generated data as accurately as possible. The G network, on the other hand, has been trained to deceive the D network, thereby minimizing the following function [24]:

$$\log(1 - D\left(G_{(Z)}\right)) \tag{1}$$

The last step is to reduce the following function:

$$\frac{minmax}{GD} V(G, D) = E_{x - Pdata}[\log D(x)] + E_{Z_{PZ_{(Z)}}}[\log(1 - D(G(Z)))] \tag{2}$$

The above equation correctly distinguishes between real and artificial data. This equation has no closed-form solution and must be solved using repetitive and numerical methods [25]. GANs are depicted graphically in Figure 1.

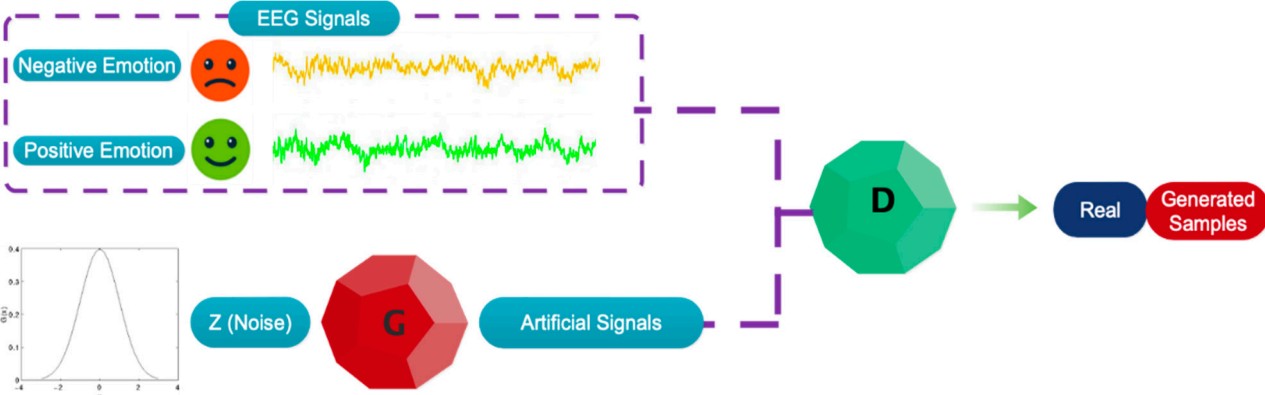

**Figure 1.** The GAN network operation.

## 2.2. Brief Description of CNN Model

CNNs have been shown to be a highly successful replacement for traditional neural networks in the development of machine learning classification algorithms. CNN learns in two stages: feed forward and reverse propagation. In general, CNN is made up of three major layers: convolutional, pooling, and fully connected (FC) layers [26]. The output of a convolutional layer is referred to as feature mapping. The max-pooling layer is typically employed after each convolutional layer that chooses just the maximum values in each feature map. A dropout layer is employed to prevent overfitting; hence, each neuron is thrown out of the network at each stage of training, with a probability. A batch normalization (BN) layer is commonly used to normalize data within a network and expedite network training. The BN is defined as follows [27]:

$$
\begin{aligned}
\widehat{u}^{(L-1)} &= \frac{u^{*(L-1)} - \mu}{\sqrt{\sigma^2 + \varepsilon}} \\
y^{*(L)} &= \gamma^{(L)} \widehat{u}^{(L-1)} + \alpha^{(L)}
\end{aligned}
\tag{3}
$$

where $u^{*(L-1)}$ is the input vector to the BN layer, $y^{*(L)}$ Y is the output vector linked to a neuron in layer $L$, $\mu = \mathrm{E}[u^{*(L-1)}]$, $\sigma^2 = \mathrm{var}[u^{*(L-1)}]$, $\varepsilon$ is a small constant for numerical stability, and $\gamma^{(L)}$ and $\alpha^{(L)}$ are the scale and shift factors learned respectively. One of the most significant components of deep neural network (DNN) is the performance of activation functions, because activation functions play an important part in the learning process. An activation function is used after each convolutional layer. Various activation functions, such as ReLU, Leaky-ReLU, ELU, and Softmax, are available to increase learning performance on DNN networks. Since the discovery of the ReLU activation function, which is presently the most-often-used activation unit, DNNs have come a long way. The ReLU activation function overcomes the gradient removal problem while simultaneously boosting learning performance. The ReLU activation function will be described as follows [27]:

$$
q(f) = \begin{cases} f & if\ f > 0 \\ 0 & \text{otherwise} \end{cases}
\tag{4}
$$

The Softmax function takes as input a vector z of K real numbers, and normalizes it into a probability distribution consisting of K probabilities proportional to the exponentials of the input numbers. That is, prior to applying Softmax, some vector components could be negative, or greater than one, and might not sum to 1, but after applying Softmax, each component will be in the interval (0,1), and the components will add up to 1, so that they can be interpreted as probabilities. Furthermore, the larger input components will

correspond to larger probabilities. The standard (unit) Softmax function $\sigma : \mathbb{R}^K \to (0,1)^K$ is defined when $K \geq 1$ by the Equation (5) [27]:

$$\sigma(z)_i = \frac{e^{z_i}}{\sum_{j=1}^{k} e^{z_j}} \text{ for } i = 1, \ldots, k, \ z = (z_1, \ldots, z_k) \in \mathbb{R}^k \tag{5}$$

A loss function is used in the prediction stage of DNN models to learn the error ratio. The loss function is a means of measuring and describing model efficiency in machine learning approaches. The error criteria is then minimized using an optimization strategy. Indeed, the optimization findings are employed to update hyperparameters [27].

*2.3. Type-2 Fuzzy Sets*

In 1975, type-2 fuzzy sets were introduced as an extension of type-1 fuzzy sets [28,29]. In contrast to type-1 fuzzy systems, belonging functions in type-2 fuzzy systems have fuzzy membership degrees. In comparison to conventional fuzzy systems (type-1 of fuzzy functions), the use of type-2 fuzzy belonging functions improves fuzzy systems' ability to deal with uncertainty (such as measurement noise). This capability of type-2 fuzzy systems has been used in research to design systems with high uncertainty and complexity (such as control systems), and its efficiency has been demonstrated in theoretical and practical applications. The type-2 fuzzy activation function has lately been used to enhance neural network efficiency [29]. As a result, the occurrence of this function in neural networks is as follows:

$$f(\sigma; \gamma) = \begin{cases} P\sigma k(\sigma), & \text{if } \sigma > 0 \\ N\sigma(-\sigma), & \text{if } \sigma \leq 0 \end{cases} \tag{6}$$

where the function $k$ can be expressed as follows [28]:

$$k(\sigma) = \frac{1}{2} \left( \frac{1}{\alpha + \sigma - \alpha\sigma} + \frac{-1 + \alpha}{-1 + \alpha\sigma} \right) \tag{7}$$

The parameters $\gamma = [\alpha, P, N]$ must be updated in each iteration, and the update algorithm is represented by the following equations:

$$\frac{\partial L}{\partial \gamma_C} = \sum_j \frac{\partial L}{\partial f_c(\sigma_{cj})} \frac{\partial f_c(\sigma_{cj})}{\partial \gamma_c} \tag{8}$$

where $c$ represents the layers, $j$ represents the observation element, and $L$ represents the DNN loss function. In addition, the $\frac{\partial L}{\partial f_c(\sigma_{cj})}$ represents the slope emitted from the deeper layers following the type-2 fuzzy activation layer, and is equal to [29]:

$$\frac{\partial f_c(\sigma_c)}{\partial a_c} = \begin{cases} \frac{p_c\sigma_c}{2}\left(\frac{1}{\alpha_c\sigma - 1} + \frac{\sigma_c - 1}{(a_c + \sigma_c - \alpha_c\sigma_c)^2} + \frac{\sigma_c(1 - a_c)}{(a_c\sigma_c - 1)^2}\right) \\ \text{if } \sigma_c > 0 \\ -\frac{N_c\sigma_c}{2}\left(\frac{1}{\alpha_c\sigma + 1} + \frac{\sigma_c + 1}{(a_c - \sigma_c + \alpha_c\sigma_c)^2} + \frac{\sigma_c(1 - a_c)}{(a_c\sigma_c + 1)^2}\right) \\ \text{if } \sigma_c \leq 0 \end{cases} \tag{9}$$

and also:

$$\begin{aligned} \frac{\partial f_c(\sigma_c)}{\partial P_C} &= \begin{cases} \sigma_c k_c(\sigma_c), & \text{if } \sigma_c > 0 \\ 0, & \text{if } \sigma_c \leq 0 \end{cases} \\ \frac{\partial f_c(\sigma_c)}{\partial N_C} &= \begin{cases} 0, & \text{if } \sigma_c > 0 \\ \sigma_c k_c(-\sigma_c), & \text{if } \sigma_c \leq 0 \end{cases} \end{aligned} \tag{10}$$

The $k_c(.)$ in the above equation is obtained from the law of updating the parameters in the following form:

$$\Delta\gamma = \rho\Delta\gamma + \xi\frac{\partial L}{\partial \gamma} \tag{11}$$

where $\rho$ and $\xi$ parameters represent the amount of movement and training rate, respectively. The total number of learnable/adjustable parameters when using the type-2 fuzzy activation function is only $3C$ ($C$ is the number of hidden units); this number is relatively small when compared to the total number of ordinary DNN weights. Because of the mentioned abilities of type-2 fuzzy sets, the belonging functions of type-2 fuzzy activation functions are used instead of the usual activation functions in the proposed model's hidden layers to deal with uncertainties, measurement noises, and improve detection accuracy [29].

## 3. Suggested Method

The details of data collection will be presented first in this section. Then, in the following section, how to prepare and pre-process the data, design the proposed architecture, and prepare the training and evaluation sets will be thoroughly explained. Figure 2 presents a graphical representation of the block diagram of the proposed model in order to automatically detect positive and negative emotions.

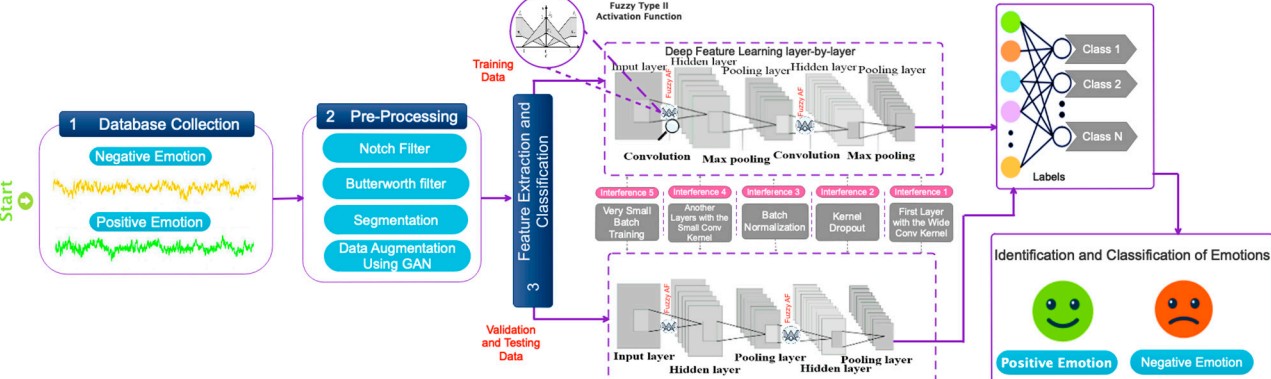

**Figure 2.** A graphical representation of the proposed model.

### 3.1. Data Collection

To create the database, EEG signals were used to recognize emotions in eleven volunteers (five men and six women) ranging in age from 18 to 32 years. All of the volunteers who participated in the experiment were free of any underlying disease. They had also read and signed the written consent form in order to participate in the experiment. The ethics committee of Tabriz University approved this experiment in the BCI laboratory with license 1398.12.1. The volunteers were instructed to abstain from alcohol, medications, caffeine, and energy drinks for 48 h prior to the test. In addition, volunteers were asked to bathe the day before the test and refrain from using hair-softening shampoos. All of the tests were completed at 9 a.m. to ensure that everyone had enough energy. Before the experiment, the Beck Depression Mood Questionnaire was used to screen out depressed volunteers. Candidates who scored higher than 21 on this test were diagnosed with depression and were removed from the testing process. The reason for using this test is that people with depressive disorders have a lack of emotional induction. Furthermore, the SAM assessment test in paper form with 9 grades was used to control the valence and arousal dimensions. A score of less than 3 and a score of more than 6 on the relevant test were considered low and high grades, respectively.

A 21-channel EEG device conforming to the 10–20 standard was used to record EEG signals. To reference brain signals, two electrodes, A1 and A2, were used. As a result, 19 of the 21 channels were actually available. Volunteers were asked to keep their eyes closed during EEG signal recording to avoid electrooculogram (EOG) signal artifacts. The sampling frequency was 250 Hz, and the electrodes had an impedance matching of 10 kΩ. One of the participants during the recording of EEG signals is shown in Figure 3.

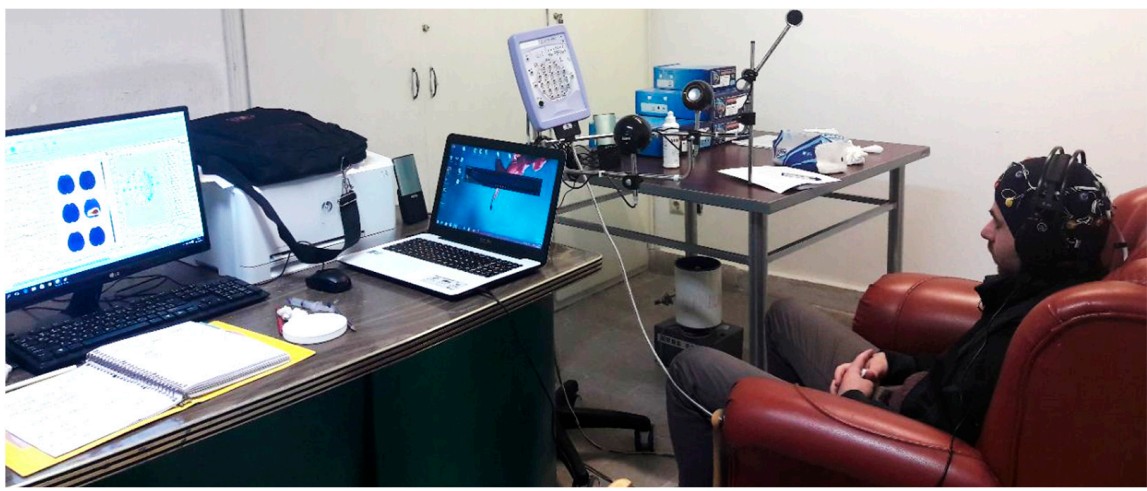

**Figure 3.** One of the participants in musical stimulation and EEG recording.

Table 1 shows the details of the experiment participants' signal recordings as well as the reasons for removing some participants from the experimental process. To clarify the reason for participant exclusion, participant 6 was excluded from the experiment, due to a low level of positive emotional arousal (6). In addition, due to a depression disorder (22 > 21), Participant 2 was barred from continuing the signal recording process.

In this study, musical stimuli were used to elicit positive and negative emotions in subjects. As a result, 10 pieces of music in happy and sad styles were played through headphones for the participants. Each piece of music was played for 30 s to the subjects, and their EEG signals were recorded. In order to prevent the transfer of produced excitement, 15 s of rest (neutral) was considered for each person between each piece of music played for people. As a result, the signals recorded for people while listening to happy and sad songs were labeled as positive and negative emotion, respectively. Finally, for each emotional state, 2.5 min of signal were recorded, and with a sampling frequency of 250, the number of samples for each positive and negative emotional state was 37,500. For the musical stimulation of the participants in the experiment, for two positive and negative emotions, in accordance with [30], a sad theme was used to induce negative emotion and an emotional (exciting) theme was used to induce positive emotion. Table 2 lists the Persian songs that were played for the subjects. Figure 4 depicts the order in which music was played for subjects.

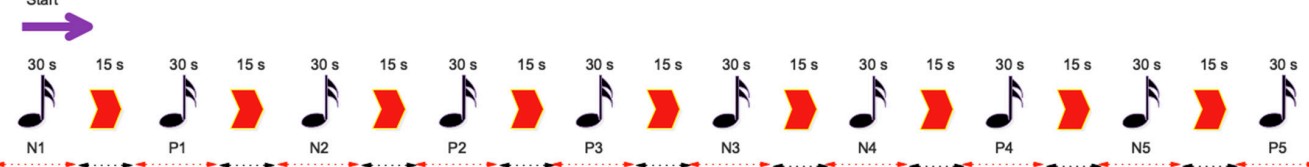

**Figure 4.** The order and time of playing positive (P) and negative (N) music for the participants.

**Table 1.** Details related to recording the signal of participants in the experiment.

| Subject | 1 | 2 | 3 | 4 | 5 | 6 | 7 | 8 | 9 | 10 | 11 | 12 | 13 | 14 | 15 | 16 |
|---|---|---|---|---|---|---|---|---|---|---|---|---|---|---|---|---|
| Sex | M | M | F | M | M | M | M | M | M | F | F | F | F | M | F | M |
| Age | 25 | 24 | 27 | 24 | 32 | 18 | 25 | 29 | 30 | 19 | 18 | 20 | 22 | 24 | 23 | 28 |
| BDI | 16 | 22 | 19 | 4 | 0 | 11 | 13 | 19 | 20 | 14 | 22 | 12 | 0 | 12 | 1 | 9 |
| Valence For P emotion | 9 | 6.8 | 6.2 | 7.4 | 5.8 | 5.6 | 7.2 | 7.8 | 7.4 | 6.8 | 7.8 | 8.6 | 6 | 8 | - | 7.4 |
| Arousal For P emotion | 9 | 6.2 | 7.4 | 7.6 | 5 | 5.4 | 7.4 | 7.4 | 7 | 6.6 | 8 | 8.6 | 6 | 8 | - | 8 |
| Valence For N emotion | 2 | 3.6 | 4.2 | 2.4 | 4.4 | 2 | 3.8 | 2.8 | 3.4 | 3.8 | 4.5 | 2 | 2 | 1.8 | - | 1.8 |
| Arousal For N emotion | 1 | 2 | 4.6 | 2.6 | 5.6 | 1.6 | 3.8 | 3 | 5.4 | 3.2 | 3 | 1.2 | 1.2 | 1.6 | - | 2 |
| Result of Test | ACC | REJ | REJ | ACC | REJ | REJ | REJ | ACC | REJ | REJ | REJ | ACC | ACC | ACC | REJ | ACC |
| Reason for rejection | - | Depressed 21 < 22 | Failure in the SAM test | - | Failure in the SAM test | Failure in the P emotion | Failure in the N emotion | - | Failure in the N emotion | Failure in the N emotion | Depressed 21 < 22 | - | - | - | Motion noise | - |

**Table 2.** The type of music and induced excitement used [30].

| Emotion Sign and Music  Number | The Type of Emotion Created in the Subject | The Name of the Music |
|---|---|---|
| N1 | Negative | Advance income of Isfahan |
| P1 | Positive | Azari 6/8 |
| N2 | Negative | Advance income of Homayoun |
| P2 | Positive | Azari 6/8 |
| P3 | Positive | Bandari 6/8 |
| N3 | Negative | Afshari piece |
| N4 | Negative | Advance income of Isfahan |
| P4 | Positive | Persian 6/8 |
| N5 | Negative | Advance income of Dashti |
| P5 | Positive | Bandari 6/8 |

*3.2. Data Pre-Processing*

As is clear, the use of all EEG channels will increase the computational load, due to the increase in the dimensions of the feature matrix. Therefore, it is necessary to identify active channels. For this purpose, in this work, 12 active electrodes were identified according to [31], and the rest of the electrodes were excluded from the processing process. The selected electrodes include Pz, T3, C3, C4, T4, F7, F3, Fz, F4, F8, Fp1 and Fp2. As is known, the data of EEG signals are highly sensitive to noise, and pre-processing must be carried out on the recorded data before the processing operation. To achieve this goal, several operations are performed on the data to prepare the data for processing. First, a notch filter is applied to the data in order to remove the 50 Hz frequency of the power supply. In the next step, considering that the effective range for emotional induction from EEG signals is between 0.05 and 45 Hz [20], a second-order Butterworth filter is applied to the data in this range. Furthermore, GAN networks are used to augment data and prevent the phenomenon of overfitting. The G network uses a random vector with a uniform distribution and dimensions of 100 to generate an output with dimensions of $37{,}500 \times 1$. G network architecture includes four fully convolutional layers with 256, 512, 1024, and 37,500 dimensions, as well as batch normalization in each layer. Also included is the ReLU activation function, which has a batch size of 10 and a learning rate of 0.0001. It has been used in the G network for 100 epochs. The D network analyzes images with dimensions of $37{,}500 \times 1$ to determine whether they are fake or real. This network employs four fully connected layers, as well as a dropout in the first layer. The sigmoid activation function is also used at the network's end. Cross-entropy indicators and the Adam optimizer are used in this network. The number of samples increased from 37,500 to 75,000 after using GAN networks, nearly doubling. Figure 5 compares the EEG signal generated by the GAN network with the real signal for two positive and negative emotions. As shown in the figure, 8 s of the real EEG signal from electrode C4 with negative and positive emotion are displayed with 8 s of the generated EEG signal by the GAN network. According to the figure, as can be seen, there is a high similarity between the real and the generated signal, which indicates that the GAN network has been effective in generating EEG data.

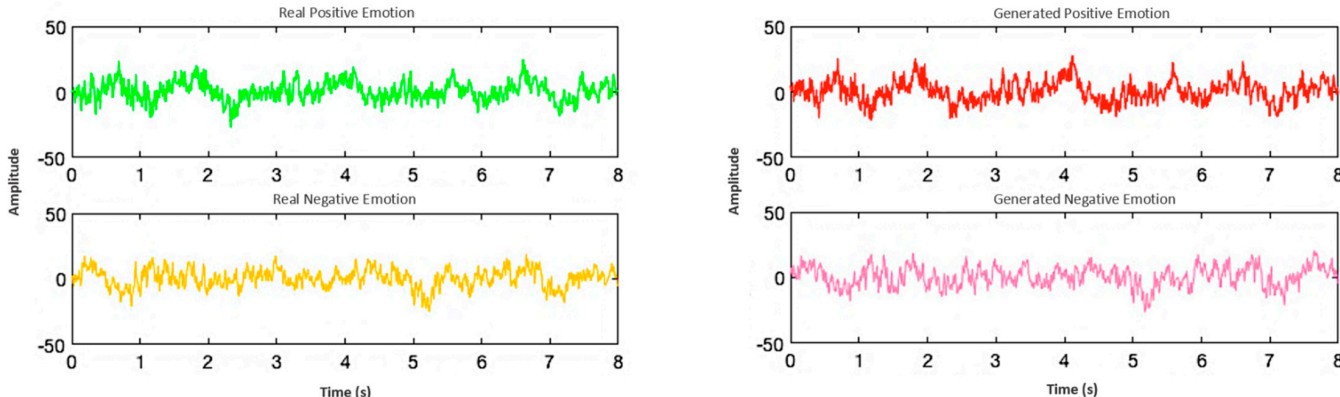

**Figure 5.** Comparing the EEG signals generated by the GAN network with the real signals.

As we know, EEG signals are non-stationary, and in order to stabilize them, these signals are considered as small epoch. In this study, in order to facilitate the data training process and prevent the overfitting problem, a segmentation technique was performed on the signal at the last stage. Thus, the signal with the length of 75,000 was divided into 540 epochs with the length of 2000 samples. Finally, according to the number of emotional classes, the number of electrodes considered and the segmentation operation, the final dimensions considered to enter the proposed network are $(7560) \times (36 \times 2000 \times 1)$.

### 3.3. Proposed Deep Architecture

For the suggested paradigm, the following were chosen or carried out:

I.      A P = 0.3 dropout layer;
II.     A convolution layer with a type-2 fuzzy activation function, followed by the max-pooling layer and the BN layer;
III.    The preceding phase is repeated three times;
IV.    A convolution layer with a type-2 fuzzy activation function, followed by a BN layer;
V.     The preceding process is carried out once more. To calculate points, the FC layer is combined with the nonlinear Softmax activation function.

Table 3 shows the proposed model's number of filters, stride size, and architectural details. The dimensionality of the hidden layers is reduced from input size to 2 in the proposed model (number of classes). Figure 6 depicts the architectural details of the proposed model. It should be noted that the hyperparameter values have been adjusted, based on a review of related works and tests. Finally, the best hyperparameters are chosen for the proposed model. The momentum and gamma parameters in the BN layer are set to 0.8, and the weight decay rate is set to $5 \times 10^{-5}$. Most functions are investigated as optimizers, including Stochastic Gradian Descend (SGD) [32], Adam [32], StepLR [33], CyclicLR [33], and ReduceLR [33]. However, because the proposed model outperforms the SGD algorithm in practice, it is used as the optimizer with a learning rate of 0.0001 and batch size 64. Furthermore, the cross-entropy loss function [34] controls the training process. The optimal hyperparameters for the proposed model are shown in Table 4.

The training, evaluation, and validation sets are arranged in the following order, to evaluate the proposed model: from the total available data of EEG signals, 70% of the data (5292 samples) for the training set, 10% of the data (756 samples) for the validation set, and 20% of the data (2268 samples) for the test set were chosen. Figure 7 depicts a graphical representation of this division.

**Table 3.** Details of the layers in the construction of the proposed network.

| L | Layer Type | Activation Function | Output Shape | Size of Kernel and Pooling | Strides | Number of Filters | Padding |
|---|---|---|---|---|---|---|---|
| 0–1 | Convolution 2-D | Fuzzy | (None, 18, 1000, 16) | 128 × 128 | 2 | 16 | **yes** |
| 1–2 | Max-Pooling 2-D | - | (None, 9, 500, 16) | 2 × 2 | 2 | - | **no** |
| 2–3 | Convolution 2-D | Fuzzy | (None, 9, 500, 32) | 3 × 3 | 1 | 32 | **yes** |
| 3–4 | Max-Pooling 2-D | - | (None, 4, 250, 32) | 2 × 2 | 2 | - | **no** |
| 4–5 | Convolution 2-D | Fuzzy | (None, 4, 250, 32) | 3 × 3 | 1 | 32 | **yes** |
| 5–6 | Max-Pooling 2-D | - | (None, 2, 125, 32) | 2 × 2 | 2 | - | **no** |
| 6–7 | Convolution 2-D | Fuzzy | (None, 2, 125, 32) | 3 × 3 | 1 | 32 | **yes** |
| 7–8 | Max-Pooling 2-D | - | (None, 1, 62, 32) | 2 × 2 | 2 | - | **no** |
| 8–9 | Convolution 2-D | Fuzzy | (None, 1, 62, 32) | 3 × 3 | 1 | 32 | **yes** |
| 10–11 | Convolution 2-D | Fuzzy | (None, 1, 62, 16) | 3 × 3 | 1 | 16 | **yes** |
| 11–12 | Flatten | - | (None, 100) | - | - | - | **-** |
| 12–13 | Softmax | - | (None, 2) | - | - | - | **-** |

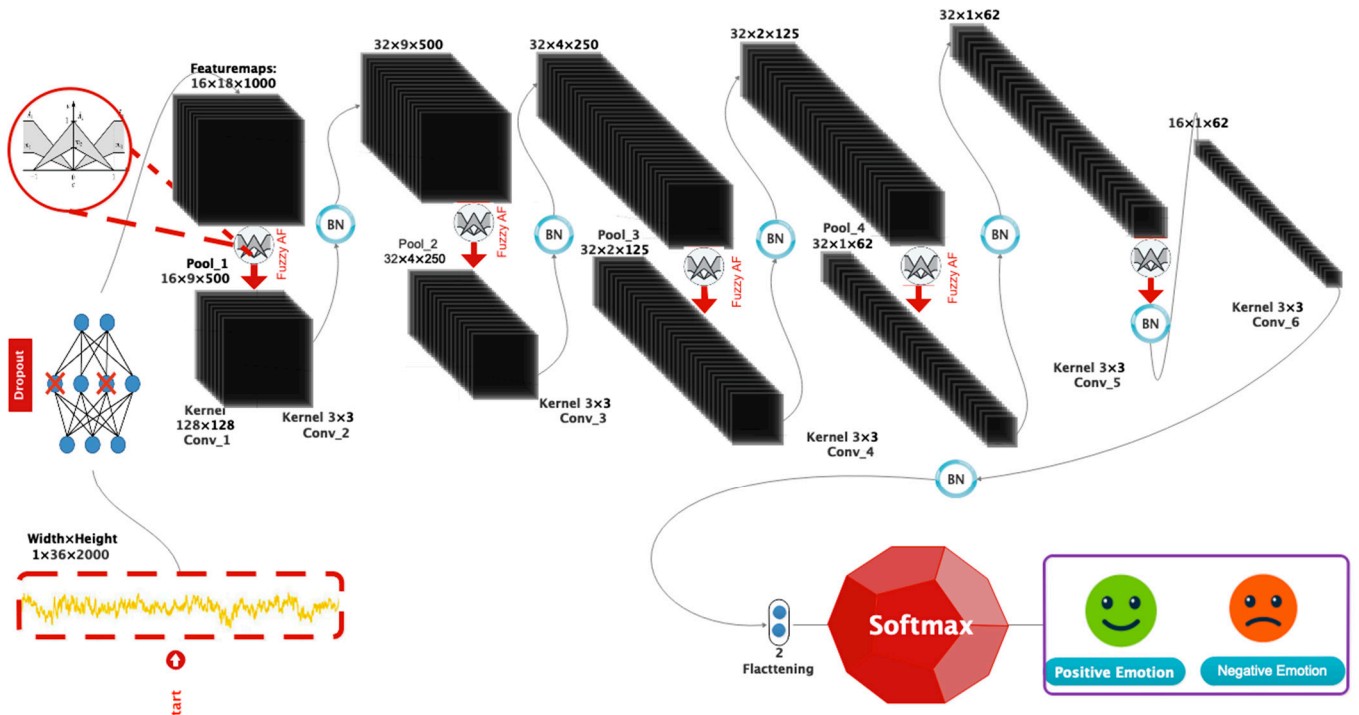

**Figure 6.** A graphical representation of the layers considered in the proposed architecture.

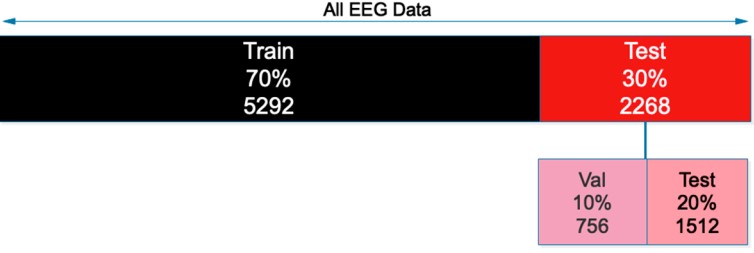

**Figure 7.** How to allocate data for training and test sets.

**Table 4.** Parameters tested in the proposed architecture.

| Parameters | Search Space | Optimal Value |
|---|---|---|
| Optimizer | RMSProp, Adam, Sgd, Adamax, Adadelta | SGD |
| Cost function | MSE, Cross-entropy | Cross-Entropy |
| Number of Convolution layers | 3, 5, 6, 9, 11 | 6 |
| Number of FC layers | 2, 3, 5 | 1 |
| Number of Filters in the first convolution layer | 16, 32, 64, 128 | 16 |
| Number of Filters in the second convolution layer | 16, 32, 64, 128 | 32 |
| Number of Filters in another convolution layers | 16, 32, 64, 128 | 32 |
| Activation Function | Relu, Leaky Relu, Type II Fuzzy | Type II Fuzzy |
| Size of filter in the first convolution layer | 3, 16, 32, 64, 128 | (128, 128) |
| Size of filter in another convolution layers | 3, 16, 32, 64, 128 | (3, 3) |
| Dropout rate | 0, 0.2, 0.3, 0.4, 0.5 | 0.3 |
| Batch size | 4, 8, 10, 16, 32, 64 | 64 |
| Learning rate | 0.01, 0.001, 0.0001 | 0.0001 |

## 4. Results

In this section, the proposed model's results will be presented and compared to previous studies. All relevant simulations were run on a computer system equipped with a Core i9 processor, 32 GB of RAM, and a GPU 2080.

Figure 8 depicts the accuracy and error of the proposed method for automatic recognition of two positive and negative emotions for training and validation sets in 200 iterations of the algorithm, which is based on the combination of convolutional and type-2 fuzzy networks. As is well known, the proposed method's correctness was demonstrated in 80 iterations of the algorithm and reached 98%. In addition, the network error was reduced from 1.8 to 0.001. Figure 9 depicts the receiver operating characteristic (ROC) analysis and confusion matrix for automatic recognition of positive and negative emotions. According to this diagram (Figure 9a), the optimal placement of the curves for both positive and negative emotions is between 0.9 and 1, indicating optimal performance in the proposed method's classification process. Furthermore, according to the confusion matrix (Figure 9b), only 18 samples of positive emotion and 9 samples of negative emotion were incorrectly recognized, indicating that the proposed network was very efficient in separating samples of each class. Figure 10 depicts the t-distributed stochastic neighbor embedding (t-SNE) graph for two emotion classes for different network layers. As shown, almost all samples of the two classes are completely separated from each other in the network's final layer.

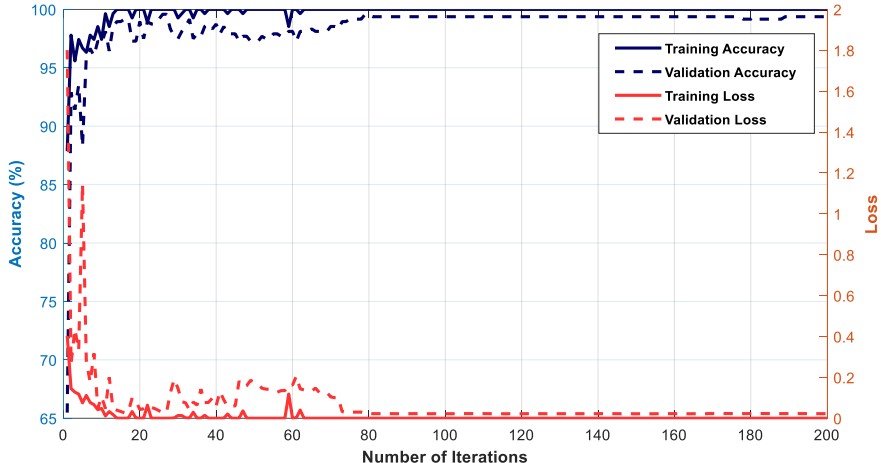

**Figure 8.** Performance of the proposed model.

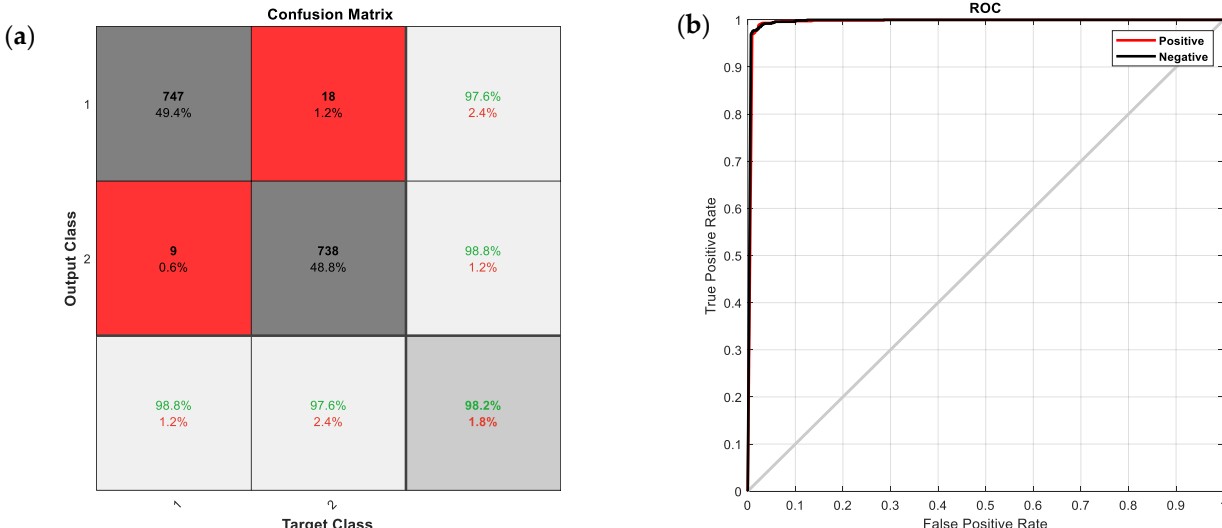

**Figure 9.** ROC analysis (**a**) and confusion matrix (**b**) based on the proposed model.

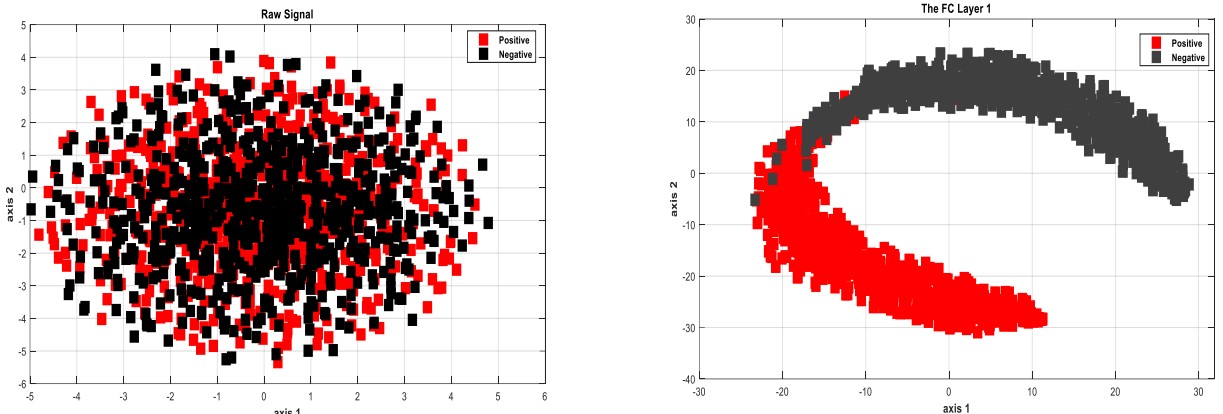

**Figure 10.** Visual representation of examples for two different layers of the proposed network.

We used and compared the two popular activation functions ReLU and Leaky-ReLU in the proposed deep convolutional architecture to determine the importance of using the combination of type-2 fuzzy functions in the CNN architecture. Figure 11 depicts the obtained results. As is well known, the use of type-2 fuzzy functions in conjunction with deep convolutional architecture has increased the accuracy of the network and allowed it to converge to the expected value faster than the ReLU and Leaky ReLU functions. In addition, as can be seen, network error is reduced in this mode. The significance of using type- 2 fuzzy functions in the proposed architecture is thus determined. Figure 12 depicts a bar chart of evaluation indices such as accuracy, precision, kappa, sensitivity, and specificity for the convolutional network, using ReLU, Leaky-ReLU, and type-2 fuzzy functions. As is well known, the obtained results for all evaluation indicators of the proposed model are greater than 95%, indicating that the proposed network is efficient. Table 5 displays the deep network training time for various ReLU, Leaky-ReLU, and type-2 fuzzy functions. As can be seen, training time for type-2 fuzzy functions is slightly longer than for the other two functions compared. This shortcoming, however, can be compensated for by using GPU processors.

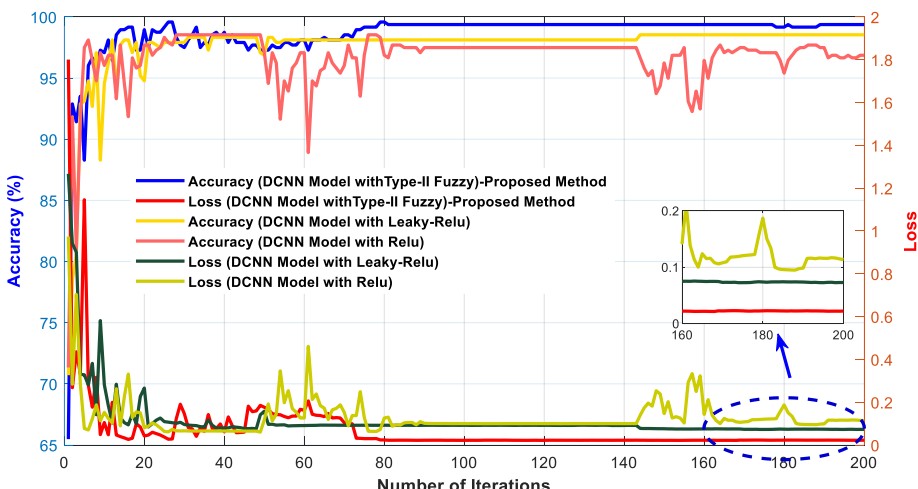

**Figure 11.** Comparing the accuracy and error of the proposed deep network with different functions.

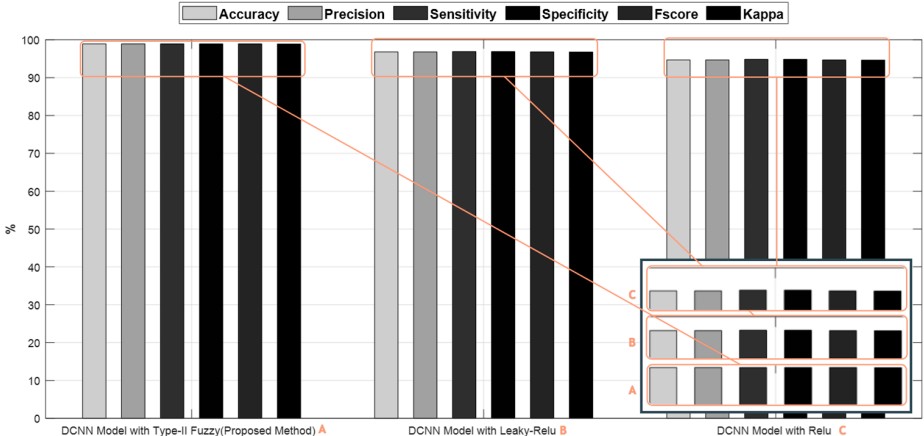

**Figure 12.** Bar graph of the proposed deep network with different functions ((**A**) type II fuzzy, (**B**) Leaky-Relu and (**C**) Relu).

**Table 5.** Comparison of network training time with different functions.

| Function Used | Relu | Leaky-Relu | Type-2 Fuzzy |
|---|---|---|---|
| **Training Time** | 10,800 s | 10,978 s | **13,571 s** |

Furthermore, the proposed network was tested in a simulated noisy environment for further evaluation. To that end, Gaussian white noise was added to the recorded EEG data at various SNRs, and the performance of the proposed network was compared to that of other activation functions such as type-2 fuzzy, ReLU, and Leaky-ReLU. Figure 13 depicts the obtained results. As can be seen, using type-2 fuzzy functions with deep convolutional networks (the proposed method) keeps high accuracy above 90% across a wide range of SNRs. Given the sensitivity of EEG signals to noise, the network used for emotion classification must be noise-resistant, in order to be used in real-time applications. As can be seen, the proposed network can ensure high classification accuracy in noisy environments and can be used in BCI applications. The three factors listed below explain why the proposed network is resistant to environmental noise:

(a)    Using large filters in the first layer.
(b)    Using small filters in the middle layer.
(c)    Using type-2 fuzzy functions as the activation function of a convolutional layer.

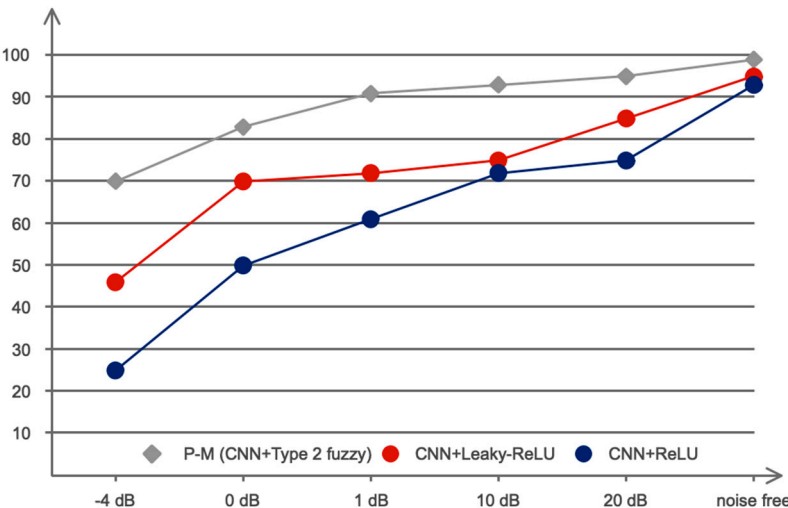

**Figure 13.** The resistance of different activation functions in the proposed network in different SNRs.

Table 6 compares and evaluates the proposed model with other previous studies in order to determine the effectiveness of the proposed method. As can be seen, the proposed method for emotion recognition has a classification accuracy of around 98.2%, whereas studies [23] and [20] have a classification accuracy of around 95.5% and 96%, respectively. However, this comparison does not appear to be fair, because the databases are not the same. As a result, the proposed model must be evaluated using current methods and algorithms based on the proposed database. For this purpose, the proposed network was compared to popular algorithms such as SVM, MLP, and CNN, using the proposed database and manual feature-extraction and feature-learning methods. The kernel of the SVM was created using a Gaussian radial basis function (RBF). The network search method was also used to optimize kernel parameters. The number of hidden layers and the training rate for the MLP network were set to 3 and 0.001, respectively. For the CNN network, the proposed network architecture (Table 3) was considered without the use of type-2 fuzzy functions and was based on Leaky-ReLU activation functions. Popular features such as mean, variance, skewness, kurtosis, and standard deviation were extracted from EEG signals, and classified based on the investigated networks using the manual feature extraction method. For the feature learning method, the raw EEG signals were fed into the networks to be compared. Table 7 compares the proposed method's results with the compared networks, as well as different manual-feature-extraction and feature-learning methods. Deep networks, as is well known, do not perform well in manual feature extraction. However, these networks can be trained with raw data and can extract useful features from the signal.

**Table 6.** Comparing the performance of recent studies with the proposed model.

| Recent Studies | Stimuli | Methods | Number of Emotions Considered | Accuracy (%) |
|---|---|---|---|---|
| Bhatti et al. [35] | Music | WT and MLP | 4 | 78.11 |
| Chanel et al. [36] | Video Games | Frequency band extraction | 3 | 63 |
| Jirayucharoensak et al. [37] | Video Clip | Principal component analysis | 3 | 49.52 |
| Er et al. [21] | Music | VGG16 | 4 | 74 |
| Sheykhivand et al. [20] | Music | CNN-LSTM | 2 | 96 |
| Hou et al. [23] | Video Clip | FPN + SVM | 3 | 95.50 |
| **Proposed model** | Music | CNN + Type-2 fuzzy | 2 | **98.2** |

**Table 7.** Comparing the performance of different models with different learning methods.

| Hand-Crafted Features | Feature Learning | Model |
|:---:|:---:|:---:|
| 81.9% | 76.1% | MLP |
| 70% | 93.8% | CNN |
| 87.4% | 78% | SVM |
| **76.6%** | **98.2%** | Proposed Model |

To determine the optimal performance of the designed architecture, the proposed network was compared to other pre-trained deep networks, such as VGG [38], ResNet [39], and Xception [40], using EEG recorded data. In recent years, compared networks have been widely used to classify data in a variety of fields. In these networks, the basic architecture was used, with optimizations based on the proposed database. Figure 14 depicts the accuracy obtained after 150 iterations. As can be seen, the proposed network outperforms these networks in terms of accuracy, and is better suited for emotion classification.

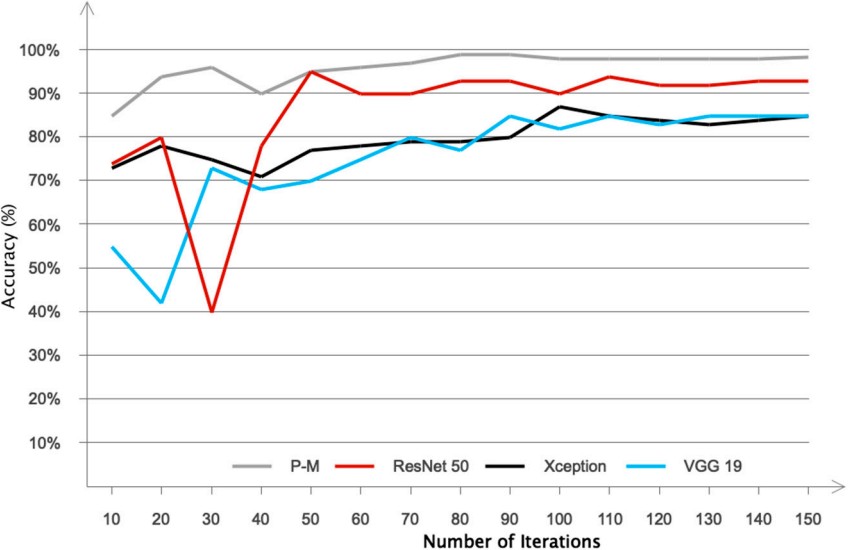

**Figure 14.** Proposed network accuracy versus popular pre-trained networks.

Despite the proposed network's favorable performance, this study, has limitations, as do others. More emotional states, such as anger, disgust, and so on, can be considered for classification in future studies, and the number of classification classes can be increased. In addition, instead of GAN networks, classical data augmentation can be used and its performance compared to the proposed method.

## 5. Conclusions

This study proposes a new method for automatically detecting positive and negative emotions that is based on the combination of type-2 fuzzy and deep convolutional networks. A standard database of EEG signals based on musical stimulation was compiled for this purpose. The proposed network architecture was then built using six convolutional layers and one softmax layer. In order to overcome uncertainties, type-2 fuzzy functions were used as activation functions in CNN architecture. The results show that the proposed method is 98.2% accurate at distinguishing between positive and negative emotions. In comparison to recent studies and methods, the proposed method performed well. Furthermore, the proposed model's performance in noisy environments was evaluated, and it was found to be resistant to a wide range of environmental noises. Given its promising performance, the proposed method can be used for a variety of applications including BCI, identity detection, and lie detection.

**Author Contributions:** Conceptualization, F.B.; methodology, A.F. and S.D.; software, S.S. and S.D.; validation, F.B. and A.F.; writing—original draft preparation, S.D. and S.S. All authors have read and agreed to the published version of the manuscript.

**Funding:** This research received no external funding.

**Data Availability Statement:** The data related to this article is publicly available on the GitHub platform under the title Baradaran emotion dataset.

**Conflicts of Interest:** The authors declare no conflict of interest.

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
