# Peer review of "Automatic Emotion Recognition from EEG Signals Using a Combination of Type-2 Fuzzy and Deep Convolutional Networks"

_electronics, doi:10.3390/electronics12102216_

Round 1

Reviewer 1 Report

The authors propose new method for automatically detecting positive and negative emotions using deep convolutional networks. The database consists of 11 volunteers that where exposed to musical stimuli to elicit positive and negative emotions in the subjects. A 21-channel EEG device was used to record EEG signals from the subjects. In the pre-processing stage, GAN networks were used to augment data and prevent overfitting.  The proposed deep architecture consists of a convolution layer with type-2 fuzzy activation function followed by a max-pooling layer and the batch normalization layer. Results of the proposed model rendered a 98% accuracy for the classification of the two classes.   

The article is clear and relevant for the scientific community, for classification of EEG signals using deep learning techniques. The authors present an appropriate experimental design to test their hypothesis. Figures and tables are well presented and help the reader to understand and follow the proposed method and the results.

The technical parts are well sounded , some terms should be spelled complete the first time that are used in the article, as example is  include DNN in line 210, tSEN on line 411. Typos in eq. 1 and 2 should be corrected. Review sentence on lines 209 and 235  to clarify narrative to the reader. Also the explanation on the lines 361 to 364 should be reviewer to make in clearer to the reader.

Author Response

Reviewer#1:

Comments:

The authors propose new method for automatically detecting positive and negative emotions using deep convolutional networks. The database consists of 11 volunteers that where exposed to musical stimuli to elicit positive and negative emotions in the subjects. A 21-channel EEG device was used to record EEG signals from the subjects. In the pre-processing stage, GAN networks were used to augment data and prevent overfitting.  The proposed deep architecture consists of a convolution layer with type-2 fuzzy activation function followed by a max-pooling layer and the batch normalization layer. Results of the proposed model rendered a 98% accuracy for the classification of the two classes. 

  1. 1. The article is clear and relevant for the scientific community, for classification of EEG signals using deep learning techniques. The authors present an appropriate experimental design to test their hypothesis. Figures and tables are well presented and help the reader to understand and follow the proposed method and the results.
  • Thanks to the esteemed reviewer, we believe that your comments have been very useful and effective in enhancing the scientific and writing framework of the manuscript. We have considered all the comments in their entirety and made every effort to correct the manuscript in the manner suggested by the honorable reviewer.

  1. 2. The technical parts are well sounded , some terms should be spelled complete the first time that are used in the article, as example is include:
    1. a. DNN in line 210
    • The manuscript is revised based on this comment.  According to the opinion of the respected reviewer, the term DNN was written in full for the first time in the manuscript.
    1. b. tSEN on line 411.
    • The term T-Sne was written in full for the first time in the manuscript.
    1. c. Typos in eq. 1 and 2 should be corrected.
    • Both equations 1 and 2 have been modified.
    1. d. Review sentence on lines 209 and 235 to clarify narrative to the reader.
    • The sentences of lines 209 and 235 were rewritten and corrected.
    1. e. Also the explanation on the lines 361 to 364 should be reviewer to make in clearer to the reader.
    • Descriptions in lines 361 to 364 have been rewritten and modified. 

       All corrected items are highlighted in the manuscript.

Reviewer 2 Report

The authors describe a new architecture to clasiffy emotions based on EEG signals. The manuscript is generally well written an clear and the underlying concepts are correct and sound.

However I have some suggestions for the authors:

- On line 183-184 the function of the Generator is explained twice with a complete oposed meaning, please correct this and clarify. The Generator either creates noisy data or naural EEG signals.

- The reference on line 219 does not contain any information about softmax activation functions and Equation 5 should be better explained since it does not correspond to the canonical Softmax equation.

- The authors should better describe the reasoning underlying the categories associated with each musical piece, since the music piece they use are not standart to the common practice so the "positive" and "negative" category seems arbitrary.

- Figure 11 is not clear since using a bar graph when having such low differences between the functions makes impossible to see any variation.

- The authors write about using a GAN to create "in silico" data to augment their dataset. However the similarity between the generated signals and the real signals is never show neither analysed. To use artificial data in the training the authors should demostrate that the artificial signals are comparable to the real data in their frequency components and phase components at least. Also, no data about the performance of the GAN is offered.

- In line 338 the authors should provide a reference to support the statement.

Author Response

Reviewer#2:

Comments:

The authors describe a new architecture to clasiffy emotions based on EEG signals. The manuscript is generally well written an clear and the underlying concepts are correct and sound. However I have some suggestions for the authors:

  • Thanks to the esteemed reviewer, we believe that your comments have been very useful and effective in enhancing the scientific and writing framework of the manuscript. We have considered all the comments in their entirety and made every effort to correct the manuscript in the manner suggested by the honorable reviewer.

  1. 1. On line 183-184 the function of the Generator is explained twice with a complete oposed meaning, please correct this and clarify. The Generator either creates noisy data or naural EEG signals.
  • The manuscript is revised based on this comment.  Yes, the opinion of the honorable reviewer is absolutely correct. According to the reviewer's opinion, the description of how the GAN network works was rewritten and modified.

“GANs have grown in popularity in the field of deep learning over the last few years. The GANs consist of two main discriminator (D) and generator (G) networks. These two components are acting exactly opposite each other. The first network, the discriminator, is trained to distinguish between real and fake input data. The second network, the genera-tor, takes a latent noise variable z as input and tries to generate fake samples that are not recognized as fake by the discriminator. The D is in charge of telling the difference be-tween real and fake signals. A network D is trained to distinguish between original and generated data as accurately as possible. The G network, on the other hand, has been trained to deceive the D network, thereby minimizing the following function [24]“.

Which is highlighted in section 2.1, page 4 and lines 180-195.

  1. 2. The reference on line 219 does not contain any information about softmax activation functions and Equation 5 should be better explained since it does not correspond to the canonical Softmax equation.
  • The manuscript is revised based on this comment. According to the opinion of the respected reviewer, the reference related to the description of the softmax function was modified. Also, more explanation about the operation of the softmax along with its formula was added to the manuscript.

“The Softmax function takes as input a vector z of K real numbers, and normalizes it into a probability distribution consisting of K probabilities proportional to the exponentials of the input numbers. That is, prior to applying Softmax, some vector components could be negative, or greater than one; and might not sum to 1; but after applying Softmax, each component will be in the interval (0,1), and the components will add up to 1, so that they can be interpreted as probabilities. Furthermore, the larger input components will correspond to larger probabilities. The standard (unit) Softmax function is defined when by the Eq (5):

Which is highlighted in section 2.2, page 5, lines 220-228 and Ref [27].

  1. 3. The authors should better describe the reasoning underlying the categories associated with each musical piece, since the music piece they use are not standart to the common practice so the "positive" and "negative" category seems arbitrary.
  • The manuscript is revised based on this comment. Yes, the opinion of the honorable reviewer is absolutely correct. However, the selection of music for inducing positive and negative emotions has been selected based on reference [30].
  • The theme and mood of the music has a general and physiological effect and affects every person with different mental and emotional mechanisms; But the size and intensity of this effect depends on the condition of nerve cells, mental history and habituation. For the musical stimulation of the participants in the experiment, for two positive and negative emotions, according to [30], a sad theme was used to induce negative emotion and an emotional (hysterical) theme was used to induce positive emotion.

Which is highlighted in section 3.1, page 9 and lines 322-324.

  1. 4. Figure 11 is not clear since using a bar graph when having such low differences between the functions makes impossible to see any variation.
  • The manuscript is revised based on this comment. According to the reviewer's opinion, for better readability and understanding, in Figure 11 (fig 12), we magnified the range between 90 and 100 with a magnifying glass.

Which is highlighted in fig 12.

  1. 5. The authors write about using a GAN to create "in silico" data to augment their dataset. However the similarity between the generated signals and the real signals is never show neither analysed. To use artificial data in the training the authors should demostrate that the artificial signals are comparable to the real data in their frequency components and phase components at least. Also, no data about the performance of the GAN is offered.
  • The manuscript is revised based on this comment. According to the opinion of the honorable reviewer, we have compared the signals produced by the GAN network graphically. As shown in the figure below, 8 seconds of the real EEG signal from electrode C4 with negative and positive emotion are displayed with 8 seconds of the generated EEG signal by the GAN network. According to the figure, as can be seen, there is a high similarity between the real and the generated signal, which indicates that the GAN network has been effective in generating EEG data.

Which is highlighted in section 3.2, page 10 and lines 355-365.

  1. 6. In line 338 the authors should provide a reference to support the statement.
  • The manuscript is revised based on this comment. According to the opinion of the respected reviewer, the relevant reference was added for the specified part.

Which is highlighted in section 3.2, and Ref [20].

Reviewer 3 Report

The article describes an approach to quite complex problem of the emotion recognition solution. Although such studies are easy to criticize because of questionable formalizability of the topic these issues are acknowladged and well addressed.

The one contribution of the authors is the creation of the reliable dataset that can be useful in future studies of the topic.

Some minor redaction is required (for example "A The Softmax activation..." in line 219).

Author Response

Reviewer#3:

Comments:

The article describes an approach to quite complex problem of the emotion recognition solution. Although such studies are easy to criticize because of questionable formalizability of the topic these issues are acknowladged and well addressed.

  • Thanks to the esteemed reviewer, we believe that your comments have been very useful and effective in enhancing the scientific and writing framework of the manuscript. We have considered all the comments in their entirety and made every effort to correct the manuscript in the manner suggested by the honorable reviewer.

  1. 1. The one contribution of the authors is the creation of the reliable dataset that can be useful in future studies of the topic.
  • Yes, due to the fact that there was no similar EEG data set for emotional induction based on musical stimulation, in this research a standard specific data set of 11 participants has been collected for automatic detection of two positive and negative emotions. All the scenarios designed for data collection are specific and have not been implemented before in any of the previous researches. For this reason, the present research is of special importance.

  1. 2. Some minor redaction is required (for example "A The Softmax activation..." in line 219).
  • The manuscript is revised based on this comment. The manuscript is revised based on this comment. According to the opinion of the respected reviewer, the reference related to the description of the softmax function was modified. Also, more explanation about the operation of the softmax along with its formula was added to the manuscript.

“The Softmax function takes as input a vector z of K real numbers, and normalizes it into a probability distribution consisting of K probabilities proportional to the exponentials of the input numbers. That is, prior to applying Softmax, some vector components could be negative, or greater than one; and might not sum to 1; but after applying Softmax, each component will be in the interval (0,1), and the components will add up to 1, so that they can be interpreted as probabilities. Furthermore, the larger input components will correspond to larger probabilities. The standard (unit) Softmax function is defined when by the Eq (5):

Which is highlighted in section 2.2, page 5 and lines 220-228.

Reviewer 4 Report

i think that paper is good, only need to improve the english and to omit the acronymun in the abstract

the rest of the paper is good

Author Response

Reviewer#4:

Comments:

I think that paper is good, only need to improve the english and to omit the acronymun in the abstract

The rest of the paper is good

  • Thanks to the esteemed reviewer, we believe that your comments have been very useful and effective in enhancing the scientific and writing framework of the manuscript. We have considered all the comments in their entirety and made every effort to correct the manuscript in the manner suggested by the honorable reviewer.
  • According to the opinion of the respected reviewer, the abbreviation was removed from the abstract. Also, the manuscript was rechecked and corrected for grammar.
